# Commonalities among dental patient-reported outcomes (dPROs)—A Delphi consensus study

**Phonsuda Chanthavisouk**[1]*, **Mike T. John**[2,3], **Danna Paulson**[4], **Swaha Pattanaik**[2]

**1** Division of Dental Therapy, Department of Primary Dental Care, School of Dentistry, University of Minnesota, Minneapolis, Minnesota, United States of America, **2** Division of Oral Medicine, Diagnosis and Radiology, Department of Diagnostic and Biological Sciences, School of Dentistry, University of Minnesota, Minneapolis, Minnesota, United States of America, **3** Division of Epidemiology & Community Health, School of Public Health, University of Minnesota, Minneapolis, Minnesota, United States of America, **4** Division of Dental Hygiene, Department of Primary Dental Care, School of Dentistry, University of Minnesota, Minneapolis, Minnesota, United States of America

* chant076@umn.edu

**Data Availability Statement:** Data has been uploaded to the University of Minnesota public data repository Link: https://doi.org/10.13020/7s0c-wv21.

## Abstract

Improvement of patients' oral health-related quality of life (OHRQoL) is the main goal of oral health care professionals. However, OHRQoL is not a homogenous construct and how to assess it is challenging because of the large number of currently available instruments. Investigating available instruments and what they have in common would be necessary for consolidation and standardization of these instruments into a smaller set of tools. If the OHRQoL dimensions including *Oral Function*, *Orofacial Pain*, *Orofacial Appearance*, and *Psychosocial Impact* are the fundamental building blocks of the dental patient's oral health experience, then these dimensions should be measured by generic multi-item dPROMs. In this study, a panel of 11 international dentists use the Delphi consensus process to determine how well 20 of these instruments measured the four OHRQoL dimensions. All 20 dPROMs questionnaires assessed at least one OHRQoL dimension while all four OHRQoL dimensions were measured by at least one dPROM instrument, i.e., the four OHRQoL dimensions were essential components of the patient's oral health experience. This shows that the currently available generic multi-item dPROMs have a lot in common, in that they share *Oral Function*, *Orofacial Pain*, *Orofacial Appearance*, and *Psychosocial Impact* as targeted dimensions. Based on these commonalities, it is plausible and desirable to move towards a single four-dimensional metric to assess oral health impact in all clinical, community-based, and research settings. This step is necessary to advance evidence-based dentistry and value-based oral health care.

## Background

Dental patient-reported outcomes (dPROs) represent what is important to dental patients [1,2]. Therefore, dental patient-reported outcome measures (dPROMs) determine which

**Funding:** The research reported in this publication was supported by the National Institute of Dental and Craniofacial Research of the National Institutes of Health, USA, under the Award Numbers R01DE022331 and R01DE028059.

**Competing interests:** The authors have declared that no competing interests exist.

dental treatments are the most effective in addressing dental patients' oral health problems. Two systematic reviews identified 155 multi-item dPROMs that capture oral disease impact [3,4]. These dPROMs provide a multitude of opportunities to measure oral disease impact on the quality of life, but they also present challenges to dentists and researchers such as which instrument to select and how to interpret instrument scores. The dPROMs essentially measure one or more of the four dimensions of OHRQoL *Oral Function*, *Orofacial Pain*, *Orofacial Appearance*, and *Psychosocial Impact* [5]. A search for commonalities among the currently available dPROMs could simplify and consolidate the complex array of existing dPROMs into a smaller and better set of instruments to measure the impact of oral disease. This would also facilitate the determination of treatment effectiveness in dental interventions comparable to the results of research studies. Consequently, evidence-based dental practice and value-based oral health care would greatly benefit from a standardization of patient-reported outcomes assessment.

A Delphi technique is such a structured process to derive a consensus about a topic such as the relationship between dPROMs and OHRQoL dimensions. The conceptual advantages of a Delphi process are accompanied by technical advantages of involving many dental experts with international representation and reasonable burden using electronic communication. This study implemented the Delphi process using a panel of international dental experts and aimed to investigate the commonalities among 20 generic multi-item dental patient-reported outcome measures (dPROMs).

## Methods and material

In the present study, 20 questionnaires that measure oral disease impact were assigned to the four OHRQoL dimensions *Oral Function*, *Orofacial Pain*, *Orofacial Appearance*, or *Psychosocial Impact* by dental experts using a Delphi process after undergoing a reliability assessment and reaching consensus. The oral disease-generic dPROMs were identified as instruments that measured oral disease impact across of a broad range of patients [3]. The 20 questionnaires contained 36 unique dPROs. These dPROs were measured by 53 dPROMs [3].

### Dental experts

The dental experts comprises of a group of 11 international dentists, known to one of the authors (MTJ). The selection criteria for dental experts were that they had to be a dentist, had to have practiced dentistry in the past year, and were fluent in the English language. Each dental expert received an invitation by email with detailed study instructions. The dental experts were requested to familiarize themselves with the contents of 20 dPROMs. They were provided the abstract of the published dPROM questionnaires by each author(s), a brief description of what the dPROM intended to measure, the reference of the dPROM, as well as the dPROM items. All dental experts who were approached to participate in the study had accepted the invitation, providing a response rate of 100%.

All dentists were full-time dentists, six were women and five men. Of those dental experts, six were from Europe and the rest from the United States. Five of the dentists were Doctorates of Dental Surgery (DDS) and a Doctor of Philosophy (PhD). The dental experts have had between five to 25 years of experience in the dental field. The major criterion for the dental experts were that they had to be a dentist. Dental experts should have had some understanding and experience about patients' functional, pain-related, aesthetical, and psychosocial problems in regards to oral health [6].

## The Delphi process

**Calibration of dental experts.** The definitions of the four dimensions of OHRQoL: *Oral Function*, *Orofacial Pain*, *Orofacial Appearance* and *Psychosocial Impact* were provided to all dental experts. They were asked to familiarize themselves with their meaning and content, even if they had an intuitive understanding of what these dimensions were. These dimensions were identified from 10,778 49-item Oral Health Impact Profile (OHIP-49) data of prosthodontic patients and general population subjects covering an age range of 40 years or more in 35 studies conducted in six countries [7–10]. The OHIP-49 data formed an aggregate secondary data set used for the factor analyses in the Dimension of Oral Health-Related Quality of Life Project [7].

**Assignment of PROM to OHRQoL dimensions using the Delphi technique.** The data was collected using an electronic survey in the English language. The survey was generated using Qualtrics software. Dental experts received an electronic link to the survey and provided their anonymous responses online by choosing a 7-digit number that was not to be shared with other dental experts or the study organizer. This number was used to link participants' responses to the corresponding dental expert in both round one and round two. While the number identified a particular dental expert, completion of all ratings was anonymous.

The dental experts were requested to assess the 20 questionnaires one by one. The dental experts were asked to indicate with the four scores as to what extent the specific questionnaire assesses each of the four dimensions. The scores could range from 0 to 10 points, where 0 points meant the questionnaire does not measure an OHRQoL dimension at all and 10 points meant the questionnaire measures the dimension perfectly.

In subsequent rounds of the Delphi process, dental experts did not have access to their own previous ratings but were shown summarized group scores from other dental experts. For each questionnaire, rating medians and interquartile for all four dimensions were presented. The dental experts were asked to provide ratings again, but this time based on the knowledge of what the group's assessment was. The dental experts were informed that group ratings can but do not have to influence their own new assessment. The Delphi process was stopped when a consensus was reached, which has been defined in the following section labelled as "Consensus—Validity assessment of panel expert ratings."

**Reliability assessment of dental experts' ratings.** To find the commonalities among dPROs using dental experts' opinions, the expert ratings needed to be stable over time, i.e., an expert should have the same opinion when asked again. Therefore, a test-retest assessment was performed by asking dental experts twice within two weeks to assign questionnaires to dimensions without knowledge of previous results. Ratings from the first round of the Delphi process were used as "test" results. Before dental experts were given feedback on the group's assessment and ask to continue with the second round of the Delphi process, another survey round ("retest") was performed to determine test-retest reliability.

Out of 11 dental experts, only seven provided retest data. We calculated an intraclass correlation coefficient (ICC) according to Fleiss [11], using a one-way ANOVA for each expert and for each dimension, resulting in 28 ICCs from 7 (number of raters) times 4 (number of dimensions). Box plots were used to present central tendency and spread of ICCs overall and per OHRQoL dimension. The median ICCs were interpreted according to Fleiss [11]. An ICC <0.40 is considered "poor," 0.40–0.75 represents "fair-to-good," and >0.75 is seen as "excellent" reliability. The software that was used to interpret this data is called, Stata.

The internal consistency of the panel expert ratings was also determined. Cronbach's alpha, the overall and by dimension, was computed for the dental expert responses when they were

asked which dimension a dPROM belonged to. A Cronbach's alpha value of 0.90 or higher was adopted as the threshold for sufficient panel expert ratings' internal consistency [12].

**Consensus–validity assessment of panel expert ratings.** A Delphi study does not lead automatically to a consensus; it simply provides an opportunity to reach a consensus [13]. Only if dental experts agree on the topic, the result can be considered a consensus [13]. Therefore, we examined the "dimensionality" of dental expert ratings. Dimensionality refers to the number and nature of the attributes reflected in the dental experts' responses. If dental expert responses could be considered unidimensional, i.e., they would only measure one attribute, it could be assumed that the dental experts have reached a consensus on the topic of commonalities among multi-item dPROMs that measure disease impact. In the case of dental experts' response ratings being multi-dimensional, i.e., several attributes would be measured, it could be assumed that the dental experts would not have a homogenous opinion about the topic, indicating that a consensus was not reached.

To determine the dimensionality of dental expert ratings, a parallel analysis was performed suggested by Horn [14]. For 9 dental experts, this technique calculated the 9 dental expert's ratings x 9 dental expert ratings correlation matrix from the 20 PRO questionnaires x 9 dental experts dataset. This technique created a scree plot [15]. The eigenvalues of the sample correlation matrix were plotted against their position from largest to smallest (1, 2, . . ., 9). The eigenvalue points related to a straight line. The parallel analysis also created a simulated dataset with 180 observations randomly sampled from 9 independent normal variates. It calculated a 9 x 9 correlation matrix for the simulated data and extracted the 9 eigenvalues, ordering them from largest to smallest. This step was repeated k times (in our case, k = 1000). From 1000 eigenvalues at positions 1, 2, . . ., 9, medians were calculated. The 9 eigenvalue medians were related to a dashed line and overlaid on the Scree plot. Finally, the intersection of the solid (actual data) and the dashed lines (simulated data) was the cutoff for determining the number of dimensions present in the data. Panel expert ratings would be considered unidimensional when only one eigenvalue would be above the dashed line of the simulated data.

**Assignment of questionnaires to OHRQoL dimensions.** Having reached a consensus, median ratings for the group of raters were calculated for each questionnaire and for all four dimensions. Consequently, each questionnaire received four median ratings. For example, the "Psychological Impact of Dental Aesthetics Questionnaire" [16] received 0, 0, 9, and 10 points for median ratings for the dimensions *Oral Function*, *Orofacial Pain*, *Orofacial Appearance*, and *Psychosocial Impact*, respectively.

From this example, it becomes clear that situations such as 0, 0, 0, and 10 points, indicating a perfect fit for only one dimension, could have rarely occurred. It is expected that dPRO questionnaires measure several dimensions. Therefore, in addition to considering only median ratings of equal to or larger than 5 on the 0 to 10 numeric rating scale as, the highest median rating (in previous example, the 10 points) and any other median ratings within 2 points of the highest median rating (in the previous example, the 9 points) were considered indicators of the dimension. Therefore, the example of the questionnaire "Psychological Impact of Dental Aesthetics Questionnaire" measured *Psychosocial Impact* (score of 10) and *Orofacial Appearance* (score of 9) but did not measure the *Oral Function* (score of 0), or the *Orofacial Pain* (score of 0).

## Two hypotheses about the presence of commonalities among dPROMs

To interpret commonalities among the 20 questionnaires, two hypotheses were generated:

1. *All 20 dPRO questionnaires assess at least one OHRQoL dimension.*

2. *All four OHRQoL dimensions are measured by at least one questionnaire.*

Confirmation of these two hypotheses would be considered evidence for the commonalities among generic multi-item dPROMs that measure oral disease impact.

## Results

### Reliability of dental expert ratings

Overall, temporal stability of the questionnaire to OHRQoL dimension assignment was "excellent" according to Fleiss' guidelines [11]. This was indicated by a median reliability coefficient of 0.86 for all dimensions and all dental experts combined. For three of the four dimensions, the median reliability coefficient exceeded the 0.75 threshold, considered "excellent" reliability, indicative of temporal stability. For the fourth dimension *Orofacial Pain* the coefficient was 0.74 (Fig 1).

The dispersion of the reliability coefficients varied across the four dimensions. More variable responses were observed for the dimension *Psychosocial Impact* compared to the other dimensions *Oral Function*, *Orofacial Pain*, and *Orofacial Appearance*. One ICC estimate (0.20) was an outlier compared to 27 estimates that ranged from 0.56 to 0.99. A Cronbach's alpha of 0.95 was calculated for all the dental experts' ratings, and an estimate of 0.90 for all the dimensions. These findings indicated internal consistency sufficient for "clinical application." [12].

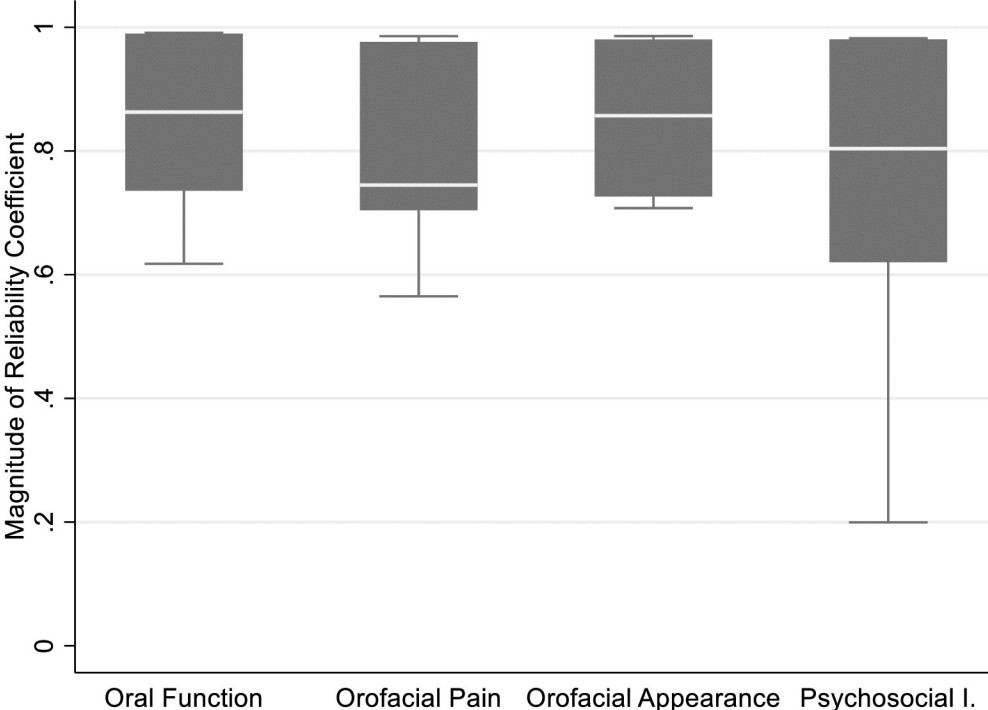

**Fig 1. Reliability coefficients (N = 28) examining the panel expert ratings' temporal stability for the four OHRQoL dimensions.**

## Validity of dental expert ratings

For all dimensions combined and for each of the four dimensions separately, a dominant general factor of uni-dimensionality among the underlying dental expert ratings was present (Fig 2). The differences among dimensions were small. The four plots showed only one eigenvalue exceeded the dashed line of the random eigenvalues, thus illustrating the uni-dimensionality of each dimension. The validity assessment findings indicated that the dental experts reached a consensus on the topic.

## Questionnaires to OHRQoL dimensions assignment

*Hypothesis 1—All 20 dPRO questionnaires assess at least one OHRQoL dimension*

Seven of the 20 dROMs measured a single dimension, seven measured two dimensions, five measure three dimensions, and one instrument measured all four dimensions, confirming the hypothesis, found in Table 1.

*Hypothesis 2—All four OHRQoL dimensions are measured by at least one questionnaire.*

*Oral Function*, *Orofacial Pain*, *Orofacial Appearance*, and *Psychosocial Impact* were measured by 14, 9, 4, and 13 questionnaires, respectively, confirming the hypothesis.

Confirmation of these two hypotheses was considered evidence for the existence of commonalities among the 20 questionnaires for adult dental patients, found in Table 1.

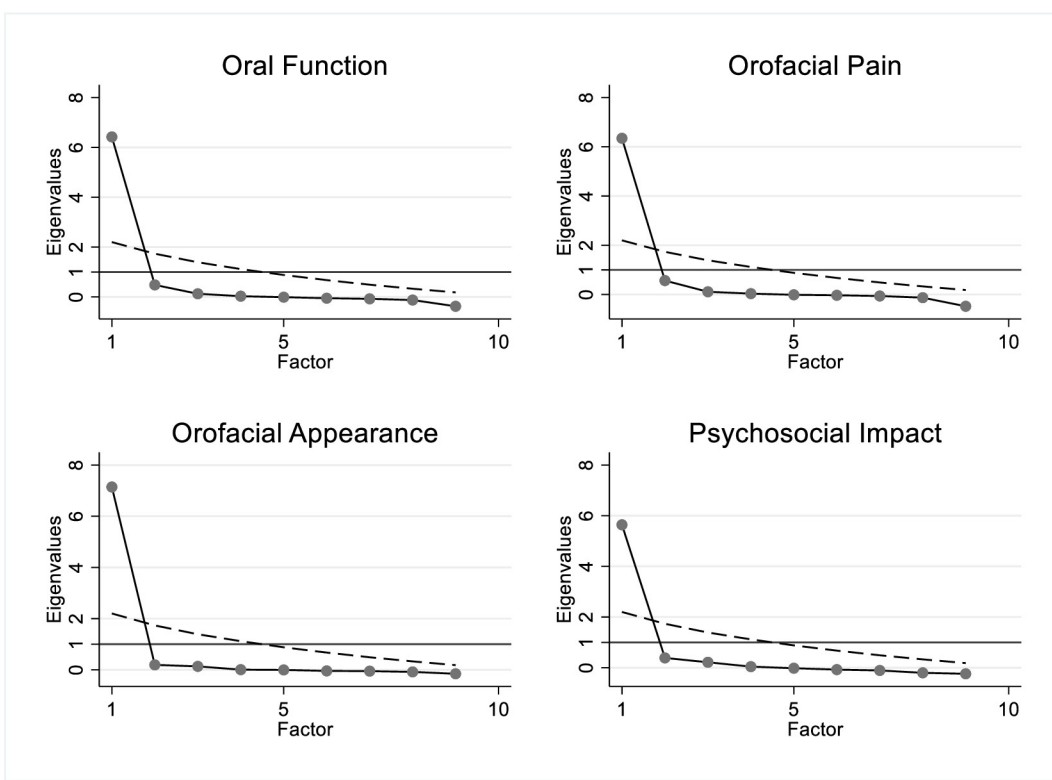

**Fig 2. Horn's parallel analysis to determine expert panelists' responses dimensionality for the four OHRQoL dimensions.**

**Table 1. dPRO questionnaires measuring OHRQoL dimensions.** "x" indicates a median expert rating ≥5 points and ≤2 points within the highest dimension rating. Empty cells indicate median expert ratings >2 points lower than the highest dimension rating.

| No. | Questionnaire | Abridged name | Authors | Publication year | Oral Function | Orofacial Pain | Orofacial Appea-rance | Psycho-social Impact |
|---|---|---|---|---|---|---|---|---|
| 1 | Rand Dental Health Index [17] | Rand DHI | Gooch, Dolan, Bourque | 1989 | | x | X | x |
| 2 | Geriatric Oral Health Assessment Index [18] | GOHAI | Atchison, Dolan | 1990 | x | x | | x |
| 3 | Jaw Disability Checklist [19] | JDC | Dworkin, LeResche | 1992 | x | | | |
| 4 | Dental Impact Profile [20] | DIP | Strauss, Hunt | 1993 | x | | | x |
| 5 | Mandibular Function Impairment Questionnaire [21] | MFIQ | Stegenga, de Bont, de Leeuw, Boering | 1993 | x | | | |
| 6 | Oral Health Impact Profile [22] | OHIP | Slade, Spencer | 1994 | x | x | | x |
| 7 | Subjective Oral Health Status Indicators [23] | SOHSI | Locker, Miller | 1994 | x | x | | x |
| 8 | Dental Impacts on Daily Living [24] | DIDL | Leao, Sheiham | 1996 | x | x | x | x |
| 9 | Oral Health Quality of Life Inventory [25] | OH-QoL | Cornell, Saunders, Paunovich, Frisch | 1997 | x | | | x |
| 10 | Oral Impacts on Daily Performance [26] | OIDP | Adulyanon, Sheiham | 1997 | x | | | x |
| 11 | Oral Health Related Quality of Life-UK [27] | OHQoL-UK | McGrath, Bedi | 2001 | x | | | x |
| 12 | Manchester Orofacial Pain Disability Scale [28] | MOPDS | Aggarwal, Lunt, Zakrzewska, Macfarlane, Macfarlane | 2005 | x | x | | x |
| 13 | Psychological Impact of Dental Aesthetics Questionnaire [16] | PIDAQ | Klages, Claus, Wehrbein, Zentner | 2006 | | | x | x |
| 14 | Jaw Functional Limitation Scale [29] | JFLS | Ohrbach, Larsson, List | 2008 | x | | | |
| 15 | Chewing Function Questionnaire—Alternative Version [30] | Alt-CFQ | Baba, John, Inukai, Aridome, Igarahsi | 2009 | x | | | |
| 16 | Modified Symptom Severity Index [31] | Mod-SSI | Nixdorf, John, Wall, Fricton, Schiffman | 2010 | | x | | |
| 17 | Orofacial Esthetic Scale [32] | OES | Larsson, John, Nilner, Bondemark, List | 2010 | | | x | |
| 18 | Brief Pain Inventory-Facial [33] | BPI- F | Lee, Chen, Urban, Hojat, Church, Xie, Farrar | 2010 | | x | | x |
| 19 | New Chewing Function Questionnaire [34] | New-CFQ | Peršić, Palac, Bunjevac, Čelebić | 2013 | x | | | |
| 20 | Craniofacial Pain and Disability Inventory [35] | CF-PDI | La Touche R, Pardo-Montero, Gil-Martínez, Paris-Alemany, Angulo-Diaz-Parreño, Suarez-Falcon, Lara-Lara, Fernandez-Carnero | 2014 | | x | | x |

## Discussion

An international panel of dental experts provided reliable and valid ratings so that a consensus could be reached in a Delphi study regarding how well 20 dental patient-reported outcome (dPRO) questionnaires fit the four dimensions of oral health-related quality of life (OHRQoL). Each questionnaire assessed at least one OHRQoL dimension, and each of the four dimensions

was at least assessed by one questionnaire. Based on these results, it was concluded that multi-item dPROMs have the four OHRQoL dimensions *Oral Function*, *Orofacial Pain*, *Orofacial Appearance*, and *Psychosocial Impact* in common.

## Comparison with literature

The commonalities among multi-item dPROMs assessing disease impact are fundamental for the meaning and scoring of questionnaires used to measure perceived oral health because they represent the major attributes measured by these instruments. While dimensionality for individual dPROMs has been studied, e.g., the Oral Esthetics Scale (OES) [36] the dimensionality of larger groups of dPROMs have not been investigated. Therefore, a direct comparison with similar studies is currently not possible.

However, previous studies which assigned names of dPROMs to the four OHRQoL dimensions are informative for interpretation of the current findings. A previous exploratory study, which investigated multi-item generic dPROMs, generated the research hypothesis for the present study [3]. In that study, the names for 53 generic dPROMs could all be assigned to the four OHRQoL dimensions. A second study which used the same methodology of relating dPROM names to the four dimensions through use of a different set of questionnaires yielded identical results [4]. In this study, all specific dPROMs were collected, i.e., instruments designed to measure impact for a specific oral disease and all (N = 102) disease-specific dPROMs could also be linked to the OHRQoL dimensions. A study investigating PROMs for pediatric dental patients came to similar conclusions [37]. In this systematic review, authors concluded the twelve pediatric dPROMs assessed included items that could be mapped to the four OHRQoL dimensions [37].

From these studies it can be concluded that, although oral disease-specific and generic dPROMs may target a different set of oral diseases in different dental patient populations like adult and pediatric, there is no difference in what they have in common as measurement targets—the four OHRQoL dimensions.

Because dPROMs capture what matters to patients, their common measurement targets should be related to the reasons why patients seek treatment for their oral health problems. An international study conducted in all six World Health Organization health regions corroborated this [37]. This study found that *Oral Function*, *Orofacial Pain*, *Orofacial Appearance*, and *Psychosocial Impact* capture dental patients' perceived impact of oral health problems worldwide, regardless of whether the patient currently suffers from oral diseases or intends to prevent them in the future [3]. While these findings applied to all dental patients, a subsequent analysis restricted to pediatric dental patients confirmed these results [37].

Although OHRQoL is only one among many concepts in the larger group of dPROs, it is the most important concept [1]. Therefore, comparing our dimensionality findings for dPROMs with those of OHRQoL instrument could provide deeper insight into dPRO dimensions. Globally, the OHIP is the most widely applied OHRQoL instrument [5]. As the most extensive OHRQoL instrument and being a special case of a dPROM, therefore it should measure the same four dimensions. Indeed, OHIP's four dimensions *Oral Function*, *Orofacial Pain*, *Orofacial Appearance*, and *Psychosocial Impact* were studied with several methodological approaches and all agreed on OHIP's four dimensions [38].

## Strengths and limitations

Methodological pros and cons of this study are centered around the basic components of the Delphi study—the expert panel, the investigated dPROMs, and methods used to achieve and demonstrate consensus.

In this present study, a small convenience sample of dental experts was selected—an approach typically adopted in Delphi studies. The participating dental experts may not represent all dentists worldwide.

Although specific dPROMs that deal with the specific aspects of oral diseases also exist, this study focused on the generic dPROMs because, conceptually, they are applicable to all oral diseases. As we studied dPROMs for adults, results may only apply to this age group. Nonetheless, the results may be applicable to older children because, the dPROMs for these children measure the same major attributes as determine by adult dPROMs.

To assess whether consensus on the commonalities among the dPROMs and OHRQoL dimensions was achieved—a necessary condition to have interpretable results—we used a method that is often applied to investigate the dimensionality of PROMs [39]. This study found strong evidence that dental experts' ratings on the commonalities of dPROMs were unidimensional, indicating that a consensus among dental experts was indeed achieved and that the Delphi study results are interpretable.

## Conclusion

*Oral Function*, *Orofacial Pain*, *Orofacial Appearance*, and *Psychosocial Impact* were solidly identified as the commonalities among generic dPROMs, supporting them as the building blocks of the dental patient's oral health experience. These findings pave the way towards a four-dimensional metric to assess oral disease impact that is essential for pragmatic implementation oral disease outcome measurement to achieve patient-centered and evidence-based clinical decision-making.

## Author Contributions

**Conceptualization:** Phonsuda Chanthavisouk, Mike T. John, Swaha Pattanaik.

**Data curation:** Mike T. John.

**Formal analysis:** Mike T. John, Swaha Pattanaik.

**Funding acquisition:** Mike T. John.

**Investigation:** Mike T. John.

**Methodology:** Mike T. John.

**Project administration:** Phonsuda Chanthavisouk, Mike T. John.

**Supervision:** Mike T. John.

**Writing – original draft:** Phonsuda Chanthavisouk, Mike T. John.

**Writing – review & editing:** Phonsuda Chanthavisouk, Mike T. John, Danna Paulson, Swaha Pattanaik.

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
