## [Decision Letter · Decision Letter 0]

3 Mar 2022

PONE-D-22-02098Commonalities among dental patient-reported outcomes (dPROs) – a Delphi consensus studyPLOS ONE

Dear Dr. Chanthavisouk,

Thank you for submitting your manuscript to PLOS ONE. After careful consideration, we feel that it has merit but does not fully meet PLOS ONE’s publication criteria as it currently stands. Therefore, we invite you to submit a revised version of the manuscript that addresses the points raised during the review process.

We look forward to receiving your revised manuscript.

Kind regards,

Kelvin Ian Afrashtehfar, M.Sc., D.D.S.,Dr. med. dent., FRCDC

Academic Editor

PLOS ONE

Journal Requirements:

[We thank Drs. Akanksha Gupta, Christian Hirsch, Daniel Reissmann, Hina Mittal, Ira Sierwald, Katrin Bekes, Oliver Schierz, Radwah Sobieh, Subha Giri, Thiago Nascimento for their help with the study. We thank Dr. Rener-Sitar for her comments to the manuscript draft.

The research reported in this publication was supported by the National Institute of Dental and Craniofacial Research of the National Institutes of Health, USA, under the Award Numbers R01DE022331 and R01DE028059.]

 [The research reported in this publication was supported by the National Institute of Dental and Craniofacial Research of the National Institutes of Health, USA, under the Award Numbers R01DE022331 and R01DE028059.]

[NO authors have competing interests.]

 This information should be included in your cover letter; we will change the online submission form on your behalf

Additional Editor Comments:

Dear Authors,

After appraising the authors’ manuscript in great detail with the effective assistance of 9 reviewers, it is my pleasure to inform you that their manuscript has been granted “ major corrections" status.

Thus, please address our suggestions and provide the revised version at your earliest convenience for a 2nd round of revisions.

Some additional literature that might support your introduction or discussion sections follows: Br Dent J. 2022 Feb;232(4):192. doi: 10.1038/s41415-022-4005-4; Evid Based Dent. 2016 Dec;17(4):109-110. doi: 10.1038/sj.ebd.6401202; Evid Based Dent. 2021 Dec;22(4):143-145. doi: 10.1038/s41432-021-0216-9; Int J Dent. 2020 Dec 29;2020:6621848. doi: 10.1155/2020/6621848.

Thank you for submitting such outstanding work to PlosOne.

Best Regards,

Academic Editor

Reviewers' comments:

Reviewer's Responses to Questions

**Comments to the Author**

1. Is the manuscript technically sound, and do the data support the conclusions?

Reviewer #1: Yes

Reviewer #2: Yes

Reviewer #3: No

Reviewer #4: Yes

Reviewer #5: Partly

Reviewer #6: Yes

Reviewer #7: Partly

Reviewer #8: Yes

Reviewer #9: Yes

2. Has the statistical analysis been performed appropriately and rigorously? 

Reviewer #1: Yes

Reviewer #2: Yes

Reviewer #3: Yes

Reviewer #4: Yes

Reviewer #5: I Don't Know

Reviewer #6: Yes

Reviewer #7: Yes

Reviewer #8: Yes

Reviewer #9: Yes

3. Have the authors made all data underlying the findings in their manuscript fully available?

Reviewer #1: Yes

Reviewer #2: No

Reviewer #3: Yes

Reviewer #4: Yes

Reviewer #5: No

Reviewer #6: Yes

Reviewer #7: Yes

Reviewer #8: Yes

Reviewer #9: Yes

4. Is the manuscript presented in an intelligible fashion and written in standard English?

Reviewer #1: Yes

Reviewer #2: Yes

Reviewer #3: Yes

Reviewer #4: No

Reviewer #5: Yes

Reviewer #6: Yes

Reviewer #7: No

Reviewer #8: Yes

Reviewer #9: Yes

5. Review Comments to the Author

Reviewer #1: This research is very interesting, in that it focuses on the quality of life. To often, we think that elimination of disease is what is important and tend to ignore the other aspects that make up good oral health. I just have a few comments in regards to the methodology. Firstly, it says 11 international dentists participated, so were only 11 dentists approached? Or were there more dentists who were approached and only 11 participated, and if so please indicate the response rate?. Secondly, it would be helpful if more information about these 11 dentists could be provided, such as their highest level of education, years of work experience ( mean - work experience of study group), full time or part time employed and where. Just to provide much more clarity.

Strengths and limitations need to be explained much more clearly. For instance, the fact that convenience sampling was employed and one of the author recruited the participants, instead of random sampling, there could be biasness in the responses.

Reviewer #2: What was the criteria for selection of the inetrnatinal experts. what was the geographical distrubution of the experts. why is no mention of the details of the selection citeria for the expert no dissussed in the text.

Reviewer #3: Comment #!

The relevance of the study was not clear.

The authors mentioned as an important factor the Delphi process using a panel of international dental experts, however the nationality of the experts was not mentioned, also the sample size of experts were small.

The methods were not concise.

Comment #2

The Future directions and the impact of the study outcomes was not clear. Importantly, the potential of this study, regarding the proposition of a new validated, clear and optimized tool for dental patient-reported outcomes was not reported.

Reviewer #4: The term ‘international data’ has been used for OHIP-49 data (ref.2), as the countries where the 35 studies were conducted represented all continents. Is it the same reason for mentioning the raters as the international dental experts i.e., did they represent all continents??? Even if that is the case, one should be cautious about using the term ‘International’ due to the prevailing diversity in different countries and within a particular country as it is related to the generalizability of the study findings.

Abstract: If we say ‘A Delphi consensus process with 11 dental experts determined how well 20 of these instruments measured the four OHRQoL dimensions, we have to present in our results as to what score on a scale of 0 to 10 had different questionnaires for a particular dimension. Actually, the aim of the study had been ‘to investigate the commonalities among 53 generic multi-item dental patient-reported outcome measures (dPROMs) used in 20 questionnaires’.

Background: The number of dPROMS investigated in one of the two systematic reviews must be rechecked. Also, the reference number assigned to various references in the reference list, the relevancy of a reference to a topic in the text, the names of authors given in Table 1 have to be reviewed.

It is suggested that the authors should collectively and thoroughly revise the manuscript for clarity, brevity, and proper sequencing of sentences and paragraphs. They are requested not to consider the amendments (in green) proposed in the attached modified version of the manuscript, as final. Some sentences (in red) in the text need to be rephrased or explained further to enhance comprehension of the manuscript.

Reviewer #5: Reviewer: This manuscript can contribute to consolidation and standardization of available instruments patients’ oral health-related quality of life.

Please see below some comments:

1. Background, page 4: In the background all references used are from articles by only a researcher.

It would be important to consider studies by others researchers.

The authors used the acronyms: dPROs; dPROMs; OHRQoL; EBDP; VBOHC

We use acronyms because its use saves space and prevent repetition. But, if the reader is not familiar with the acronym and if a paper contains too many, that can be distracting and confusing in itself. Its use will likely detract from the readability of the paper.

Then, we should be prudent in their use of abbreviations. Avoid acronyms in the unless the acronym is used multiple times in the text.

Dental patient-reported outcomes (dPROs)

dental patient-reported outcome measures (dPROMs)

Oral Function, Orofacial Pain, Orofacial Appearance, and Psychosocial

Impact (OHRQoL)

evidence-based dental practice (EBDP)

value-based oral health care (VBOHC)

2. Material and Methods, page 5:

This sentence needs a reference: “They were provided the abstract published by the questionnaire author(s), a brief description from the author on what the questionnaire intends to measure, the reference, as well as the questionnaire items.”

Material and Methods, page 5:

In this sentence: “These dimensions were identified through a systematic collection of OHIP-49 data in typical dental patients and general population subjects of both genders covering an age range of 40 years or more.”

This acronym (OHIP) lacks the initial definition.

Material and Methods, page 05 and page 06:

Where the data may be found? “Study instructions of dental experts”, the survey and results.

3. Results: the authors could bring more graphic elements to illustrate the results.

4. Discussion:

According to the authors, direct comparison with similar studies is not possible. But the authors could bring similar studies from other areas of health. Furthermore, it is noted that 15 references are citations of an only researcher.

References:

References 10, 12, 20, 24 and 25: Check references. Be consistent with referencing Vancouver style across the document.

Reference 40:

Fink-Hafner D, Dagen T, Doušak M, Novak M, Hafner-Fink M. Delphi Method: Strengths and Weaknesses. Metod Zv [Internet]. 2019;16(2):1–19. Available from:

https://www.researchgate.net/publication/337570516

Reference 40: insert link to the original source of the article:

https://mz.mf.uni-lj.si/article/view/184/287

Reviewer #6: This manuscript applied the Delphi consensus method to study commonalities among 20 pre-selected patient-reported outcomes in the dental treatment category.

The setup description and data analysis methods are sufficiently presented in the text. The Discussion of the results and the cited literature is satisfactory to the claims of the manuscript.

Nonetheless, the methods and discussion can be improved and some important points need to be clarified. Additionally, while the study satisfies its stated hypothesis, it lacks on the delphi method details and its limitations are not sufficiently presented.

1. Better clarification is needed on why only 20 questioners were chosen from a previous meta analysis by the same authors, what were the inclusion/exclusion criteria? What constituted "a generic dPROM|"? wouldn't assorted specific dPROMs for the most common dental treatment options be better and offer more power to the sample?

2, as this article builds its sample from a previous meta-analysis, a better discussion should include the methodological issues and difficulties associated specific to metal-analysis of dPROs. There is a recent publication by YUN-CHENLIU et al. on this specific issue which I recommend citing here,.

3. The Delphi method limitations and biases need to be discussed in a better, more detailed manner, as one assumes it is the main focus of this paper. additionally, more details (in the methods and discussion) need to address the characteristics of the selected experts and the issues of selection bias and group consensus vs. Individual opinion.

4. the limitations of dPROMs and how do they compare/complement Disease-oriented outcome measurements deserves to be mentioned in the intro and the in the discussion.

5. I spotted some typos and grammatical errors in the manuscript (e.g. This study implementeds... in the intro) , please re-review.

Reviewer #7: The authors aimed to " to investigate the commonalities among 20 generic multi-item dental patient-reported outcome measures (dPROMs)" using the Delphi process. However I would like to humbly suggest some editions:

1-There are a few spelling, typos and inaccuracies within the text and the references, I would suggest to review the paper;

2-In the 4th line of background it inform 153 dProms, in methods it's says 53 dProms were evaluated, am not sure if this is correct;

3-Please describe the dimensions definitions

4- Reference 1 is now published, please update

5- I'm not sure this statement would be correct "the determination of treatment

efficacy of dental interventions would greatly improve..." the questionnaires are assessing the patients' point of view, not treatment efficacy, as per your first statement "Dental patient-reported outcomes (dPROs) represent what is important to dental patients", these tools will represent what is important to the patient which not necessarily will represent what the patient need and treatment efficacy.

6-The dentists chosen to participate in the study are described as dental experts, it would be nice to have more information about this, are they experts in qualitative studies?

7-Am not sure the conclusion is precise, as stated "Oral Function, Orofacial Pain, Orofacial Appearance, and Psychosocial Impact were solidly identified as the commonalities among generic dPROMs, supporting them as the building blocks of the dental patient’s oral health experience." The four dimensionalities were the only options for the dental experts assign each questionnaire, therefore obviously these commonalities would be identified.

Reviewer #8: The presented manuscript discloses pertinent results for dental research, especially, considering the current relevance of the patient-related outcomes. The paper is properly organized and presented. Some adjustments are suggested, as follows:

1) I believe there was a mistyping error of the word “implemented” in the background section phrase “This study implementeds the Delphi process…” (page 04).

2) I understand that the “Inclusion criteria were that they had to be a dentist, had to have practiced dentistry in the past year, and were fluent in the English language” (page 04). However, did you consider and/or collect any further information on the dentists’ expertise level? Considering the possible methodological impact of the expertise variation among the participants.

3) The Fleiss reference (page 07) should be checked. I consider it should be number 16 as per the references list: “The median ICCs were interpreted according to Fleiss(5).”

4) The median ratings used for the “Assignment of questionnaires to OHRQoL Dimensions” (page 09), are based on the rate of all the experts (eleven), or only on the seven ones that provided test/re-test data? Consider clarification, as it seems relevant, especially for the abstract section methods description.

5) When mentioning a systematic review on pediatric patients (page 16), consider verifying the reference number (3) and changing it for another (37).

6) Consider checking the phrase: “Because disease-specific instruments intend to characterize the very specific impact aspects of from a particular disease, they are conceptually less suited to study the overarching themes underlying all dPROMs.” (Page 18).

Reviewer #9: The manuscript deals with a relevant topic, considered one of the dimensions of the tripod of evidence-based practice, which is the patient's perspective regarding their own oral health. A consistent survey was carried out, identifying and testing the main dimensions in 20 existing instruments. This study integrates a line of research, with other works already published by the same group. The results presented are relevant, they bring a contribution in relation to the formulated research question and the possibility of adopting a single instrument. This would allow better comparability between different studies. On the topic of discussion, the authors point out that one of the weaknesses of the study lies in the intentional sample and criteria for choosing the experts to integrate the Delphi study conducted. I agree and this seems to me to be the main weakness of the study. Even so, I suggest that the authors mention if in the criteria for choosing the specialists they included aspects such as time since graduation and practice of the profession, if they act as specialists, if they are from different specialties, if they exercise the profession inserted in different socio-cultural and economic contexts. . Perhaps these are some issues that could eventually impact the results.

6. PLOS authors have the option to publish the peer review history of their article (what does this mean?). If published, this will include your full peer review and any attached files.

Reviewer #1: **Yes: **Dilan Arun Gohil

Reviewer #2: **Yes: **Pankaj Gupta

Reviewer #3: No

Reviewer #4: **Yes: **Haleem A.

Reviewer #5: No

Reviewer #6: No

Reviewer #7: No

Reviewer #8: No

Reviewer #9: No

---

## [Author Response · Author response to Decision Letter 0]

19 Apr 2022

Thank you for taking time to review the manuscript. Please see response to reviewer document.

---

## [Decision Letter · Decision Letter 1]

9 May 2022

Commonalities among dental patient-reported outcomes (dPROs) – a Delphi consensus study

PONE-D-22-02098R1

Dear Dr. Chanthavisouk,

We’re pleased to inform you that your manuscript has been judged scientifically suitable for publication and will be formally accepted for publication once it meets all outstanding technical requirements.

Kind regards,

Kelvin I. Afrashtehfar, M.Sc., D.D.S.,Dr. med. dent., FRCDC

Academic Editor

PLOS ONE

Additional Editor Comments (optional):

Dear Respected Authors,

Your manuscript has been assessed by four reviewers this time and myself.

Fortunately enough, the process has gone smoothly without delays.

Congratulations on your acceptance.

I look forward to seeing more of your publications in PLoS One.

Regards,

The Academic Editor

Reviewers' comments:

Reviewer's Responses to Questions

**Comments to the Author**

1. If the authors have adequately addressed your comments raised in a previous round of review and you feel that this manuscript is now acceptable for publication, you may indicate that here to bypass the “Comments to the Author” section, enter your conflict of interest statement in the “Confidential to Editor” section, and submit your "Accept" recommendation.

Reviewer #1: All comments have been addressed

Reviewer #4: All comments have been addressed

Reviewer #7: All comments have been addressed

Reviewer #8: All comments have been addressed

2. Is the manuscript technically sound, and do the data support the conclusions?

Reviewer #1: Yes

Reviewer #4: Yes

Reviewer #7: Yes

Reviewer #8: Yes

3. Has the statistical analysis been performed appropriately and rigorously? 

Reviewer #1: Yes

Reviewer #4: Yes

Reviewer #7: Yes

Reviewer #8: Yes

4. Have the authors made all data underlying the findings in their manuscript fully available?

Reviewer #1: Yes

Reviewer #4: Yes

Reviewer #7: Yes

Reviewer #8: Yes

5. Is the manuscript presented in an intelligible fashion and written in standard English?

Reviewer #1: Yes

Reviewer #4: No

Reviewer #7: Yes

Reviewer #8: Yes

6. Review Comments to the Author

Reviewer #1: Authors have addressed the comments appropriately. The methodology and strengths/limitations are now more in depth. Overall, good interpretation of findings are there, and well understood.

Reviewer #4: Dear Authors

Thanks for addressing the major issues. However, the following should be looked into to rectify some minor errors.

Page 5: Data is plural as pointed out in my previos reviewed version of the manuscript.

Page 7: the statistical software Stata was used to anlyse the data, not to interpret.

Page 9: From this example, it became (not becomes) clear----you have been using past tense in this paragraph through out.

Please review thoroughly for such minor errors.

Reviewer #7: Dear authors, Thank you for taking time to review the manuscript and address the reviewers' comments/suggestions.

Reviewer #8: The authors have answered and made appropriate alterations to improve the clarity of their manuscript.

For this, I recommend its publication.

7. PLOS authors have the option to publish the peer review history of their article (what does this mean?). If published, this will include your full peer review and any attached files.

Reviewer #1: **Yes: **Dilan Arun Gohil

Reviewer #4: **Yes: **ABDUL HALEEM

Reviewer #7: No

Reviewer #8: No

---

## [Editor Report · Acceptance letter]

13 Jun 2022

PONE-D-22-02098R1 

Commonalities among dental patient-reported outcomes (dPROs) – a Delphi consensus study 

Dear Dr. Chanthavisouk:

I'm pleased to inform you that your manuscript has been deemed suitable for publication in PLOS ONE. Congratulations! Your manuscript is now with our production department. 

Kind regards, 

on behalf of

Dr. Kelvin I. Afrashtehfar 

Academic Editor

PLOS ONE